# Novel Insights into the Role of the Antioxidants in Prostate Pathology

**DOI:** 10.3390/antiox12020289

**Published:** 2023-01-27

**Authors:** Vittoria Rago, Silvia Di Agostino

**Affiliations:** 1Department of Pharmacy, Health and Nutritional Sciences, University of Calabria, 87036 Rende, Italy; 2Department of Health Sciences, Magna Græcia University of Catanzaro, 88100 Catanzaro, Italy

**Keywords:** prostate cancer, antioxidants, dietary supplement, clinical trials

## Abstract

To date, it is known that antioxidants protect cells from damage caused by oxidative stress and associated with pathological conditions. Several studies have established that inflammation is a state that anticipates the neoplastic transformation of the prostate. Although many experimental and clinical data have indicated the efficacy of antioxidants in preventing this form of cancer, the discrepant results, especially from recent large-scale randomized clinical trials, make it difficult to establish a real role for antioxidants in prostate tumor. Despite these concerns, clinical efficacy and safety data show that some antioxidants still hold promise for prostate cancer chemoprevention. Although more studies are needed, in this review, we briefly describe the most common antioxidants that have shown benefits in preclinical and clinical settings, focusing our attention on synthesizing the advances made so far in prostate cancer chemoprevention using antioxidants as interesting molecules for the challenges of future therapies.

## 1. Introduction

Prostate cancer (PCa) is the second most common type of tumor worldwide and is the fifth leading cause of cancer-related mortality among men. Despite the high morbidity for PCa, its etiology is not yet defined, the only known risk factors are age, race, and family history [1,2]. Other risk factors, such as hormones, diet, physical inactivity, obesity, smoking, sexual factors, and genetic susceptibility have been related to PCa, but the epidemiological evidence is not conclusive. Although the role of these factors is not yet defined, a complex interaction between them is likely to be associated with the development of PCa [2,3,4].

PCa is known to be linked to androgens and estrogen receptors [5]. Despite the use of prostate-specific antigen (PSA) and some recent clinical trials testing early biomarkers of PCa onset, the incidence rates of PCa have dramatically increased [4]. To date, endocrine therapy (anti-androgens combined with castration) and classical androgen deprivation (orchiectomy or luteinizing hormone-releasing hormone agonists) represent the most effective treatments for advanced and metastatic PCa. Unfortunately, although most patients favorably respond for a long time, progression to castration-resistant disease is nearly universal, and most patients eventually die from recurrent androgen-independent prostate cancer [6,7].

Increased oxidative stress is a risk factor in the onset and progression of PCa [8]. Oxidative free radicals are produced by several factors and metabolic cellular pathways, such as consumption of saturated fats and refined carbohydrates which can contribute to the development of PCa [9,10]. Oxidative damage is induced by the production of reactive oxygen species (ROS), as well as by other oxidants, such as reactive nitrogen species (RNS). Overproduction of reactive species can lead to protein/lipid oxidation and DNA mutation. The onset and progression of PCa can be prevented by modifying eating habits through modulation of the nuclear factor kappa B (NF-κB), target of rapamycin in mammals (mTOR), mitogen-activated protein kinase (MAPK), Akt, extracellular signaling regulated kinase (ERK), and phosphoinositide 3-kinase (PI3K) signaling pathways. In this context, a diet rich in vitamins A, D, and E, minerals (selenium and zinc), phytochemicals, and dietary fiber could reduce the risk of prostate cancer [11].

Because of the lack of effective drugs, early diagnosis in PCa is fundamental, since only diseases in the early stages are amenable to curative treatment, while patients with advanced disease can only be palliative. In this context, new therapeutic approaches are needed, as well as new specific biomarkers, both for early diagnosis and as tools for prognosis [12]. Several components of the diet are involved in the development and progression of PCa supporting the evidence that PCa incidence and mortality are higher in industrialized countries, where diets are generally high in fat [13]. Many dietary components may play a role in prostate development and progression cancer. Fruits and vegetables rich in antioxidants and phytochemicals have a demonstrable beneficial effect on PCa and can slow down its development and the risk of occurrence [14].

This paper aims to review the most recent data reported in the literature about the relationship between a limited number of antioxidants and prostate cancer.

## 2. Prostate Cancer

PCa is a malignant tumor of the prostate gland. This tumor tends to develop in older men, aged 50 or older, and it usually develops slowly; however, in some cases, it can be aggressive and metastasize from the prostate to other parts of the body. The human prostate is divided into three zones: peripheral, transitional, and central; in about 80% of cases, prostate adenocarcinoma originates from the peripheral or caudal area of the gland [15]. Transforming cells can be basal or luminal epithelial cells; both can result in high-grade lesions resembling adenocarcinomas [16]. Men between the ages of 50 and 75 undergo a surveillance strategy by assessing the concentration of prostate-specific antigen (PSA) in the blood, along with rectal examination to assess the size of the prostate gland. The first treatment for PCa is based on clinicopathological factors, such as PSA concentrations, clinical stage of the tumor, and histological grade, according to the Gleason score classification to the International Society of Urological Pathology Grade Group Classification [17]. After an initial diagnosis, the neoplasm is divided into low, intermediate, and high risk, and this subdivision includes multiple factors such as the number of positive biopsy specimens, tumor size, imaging results, and molecular signatures [18]. All these parameters determine the management of the disease. Generally, PCa treatment involves surgical therapy, pharmaceutical management, and androgen deprivation (ADT). ADT is frequently associated with sexual dysfunction, diabetes, cognitive dysfunction, cardiovascular disease, and alteration of the bone density [19]. Intense studies have significantly improved the management of the metastatic disease with the addition of new agents.

Several clinical trials have reported better results when androgen deprivation was combined with the use of chemotherapy drugs such as docetaxel or hormonal drugs (abiraterone, enzalutamide, or apalutamide). The introduction of Relugolix, an antagonist of AR, showed a decrease in cardiovascular side effects and suppression of serum testosterone. Poly-ADP ribose polymerase inhibitors (olaparib and rucaparib) have received Food and Drug Administration approval as they have given a significant clinical benefit in patients with deleterious mutations in the genes belonging to the homologous repair path of recombination. Recently, the combination of standard treatment with Lutetium-177 prostate-specific membrane antigen-617 improved survival in men with metastatic castration resistant prostate cancer [20].

PCa is correlated with complex interactions among inherent germline susceptibility, acquired somatic gene alterations, and micro/macroenvironmental factors. Generally, PCa presents multiple foci containing different genetic alterations with different capacity for metastatic seeding and inherent treatment resistance. Some authors argue that chronic microbial inflammation of the urinary tract participates in prostatic carcinogenesis through the generation of reactive oxygen species that induce DNA damage and selection of mutated cells. In addition, during inflammatory processes, the prostate has many proliferative luminal epithelial cells of intermediate phenotype that could be subject to epigenetic and genomic chromatin alterations, which induce malignant transformation [21].

The prostatic intraepithelial neoplasia can be categorized from low-grade to high-grade transformation. The high-grade prostatic intraepithelial neoplasia lesions can be associated with markers of transformation, such as the overexpression of the enzyme alphamethylacyl-CoA racemase (AMACR), which is associated with adenocarcinoma, the loss of the basal markers p63 (TP63), cytokeratin 5 (KRT5), and cytokeratin 14 (KRT14), and the gain of luminal markers, such as cytokeratin 8 (KRT8), while the TMPRSS2 gene is thought to be involved in luminal differentiation [15,16,21]. The most common chromosomal aberration present in more than 50% of patients with prostate cancer is represented by a gene fusion between TMPRSS2 and ERG genes [15]. However, it is thought that the epithelial transformation is secondary to a series of phenotypes and genotypic changes within a tumor-permissive inflammatory microenvironment in the prostate. More than 40% of patients showed genomic fusion of TMPRSS2–ERG, 5–15% showed loss-of-function mutations in SPOP, and 3–5% showed gain-of-function mutations in FOXA1 [15,22,23], while alterations of the androgen receptor (AR) gene are rare [22]. Furthermore, about 20% of the patients show PTEN deletions and TP53 mutations, and their frequency increases to more than 50% in cases with advanced disease. About 40% of nonmetastatic cancers have increased genetic instability, which is associated with disease recurrence [24,25]. This genomic instability, together with intratumoral hypoxia, causes highly aggressive tumors with a high probability of relapse [26]. The alterations in AR signaling should also be mentioned, which are important drivers of resistance to androgen deprivation therapy. Alterations of the AR pathway are evident in metastatic castration-resistant prostate cancer (MCRPC) [27].

Several studies have reported AR aberrations as mediators of acquired resistance toward targeting agents AR [28]. It is known that the loss of dependence on AR signaling occurs in 15–20% of advanced and treatment-resistant prostate cancers and can evolve into the castration-resistant transformation of the neuroendocrine prostate cancer. Genes involved in repairing DNA errors and breaks are also implicated in prostate cancer; in fact, men with BRCA1 or BRCA2 mutations have a higher probability of getting prostate cancer with a high incidence of disease aggression due to the additional activation of MYC in combination with inactivation of TP53 and PTEN [29,30]. About 12% of patients inherit mutations in the BRCA1, BRCA2, ATM, CHEK2, RAD51D, and PALB2 genes [31].

In particular, there is consolidated evidence from NGS data on several patient cohorts that accumulation of hotspot gain-of-function mutations in the TP53 gene can also be found at a relatively high frequency (28–36%) in primary and, especially, in naïve metastatic prostate cancer [32,33,34]. In castration-resistant prostate cancer, the TP53 mutation rate was reported between 53% and 73% [33]. From this point of view, the management of PCa is constantly evolving to try to understand the genomics and biology underlying cancer from primary to metastatic forms. Indeed, mutational perturbations could have strong potential as biomarkers to stratify the risk of the patients and to identify those who could benefit from specific treatment.

Very recently, alternative polyadenylation (APA), which is a molecular mechanism that produces mRNAs that differ in their 3′ end, has been shown to be a novel, interesting and targetable way to affect PCa carcinogenesis [35].

In this context, the development of increasingly sensitive imaging methods has significantly improved diagnostic accuracy and correct staging in order to improve surveillance strategies. The remarkable advances made by research have introduced new therapies into clinical practice, with treatments targeting genomic alterations in DNA repair pathways. A notable improvement in disease management has been achieved in the treatment of metastatic forms, with the use of several new androgen pathway inhibitors that significantly improve patient survival. Molecular typing of localized and recurrent diseases will certainly bring benefits on clinical management. Furthermore, the study of new therapies, such as targeted radioisotopes and immunotherapy, bode well for improving the lives of patients with PCa [2].

As previously remarked, among the different risk factors associated with PCa, differences in the incidence of PCa have been demonstrated as a function of diet, country of residence, and ethnicity [11,36]. These observations have promoted studies that correlate PCa with antioxidant intake from diet and supplements. To date, the effects of antioxidants on the progression of PCa are scarcely known; however, antioxidants play an important role in the body as they prevent damage from free radicals, molecules that attack healthy cells and can contribute to cancer risk.

## 3. Antioxidants

Antioxidants are substances able to counteract the production of free radicals and the oxidation process; they can be classified according to their source: endogenous sources such as enzymes, and exogenous sources such as beta-carotene, lycopene, and vitamins A, C, and E (tocopherols). The mineral element selenium is generally considered a food antioxidant, but the antioxidant effects of selenium are most likely due to the antioxidant activity of proteins that have this element as an essential component [37]. Several scientists have reported that the use of synthetic antioxidants causes health problems due to the fact that some of these compounds exhibit toxicity after their absorption, and this fact could invalidate numerous clinical studies that have been carried out on patients who have taken these supplements [37,38].

Very recently, thanks to some economic pushes and the novel therapies based on holistic medicine, the literature has reported examples of the use of plant raw materials rich in antioxidants and products derived from their processing to obtain foods fortified with antioxidants compared to their traditional formulations [39]. From this, it appears that it is very important to study the intermediate products deriving from the catabolism of the antioxidant molecules contained in foods, to have knowledge of their stability once ingested and their function as a free-radical scavenger. This step makes it possible to synthesize more functional and easily controllable active ingredients to determine a regimen while also avoiding the thermal processes that can cause modification in the chemical structure of compounds and the bioactive properties of food.

However, only a limited number of studies have been reported for vitamin E, selenium, and a few other antioxidants bioavailability [40]. For example, a very recent paper reported that hydroalcoholic pomace extracts containing high concentrations of anthocyanins, phenolic acids, flavonoids, and stilbenes had a higher antioxidant activity than aqueous extracts [41]. However, the antioxidant activity of aqueous extracts increased after intestinal digestion by promoting the proliferation of probiotic bacteria, while that of hydroalcoholic extracts dramatically decreased [41].

According to some clinical trials and experiments on in vivo models, antioxidants if not carefully administered could help the onset of cancer and interfere with chemotherapy treatments [38,42]. Furthermore, these studies often lack pharmacokinetic experiments to evaluate the presence of functionally active antioxidant molecules in the participants’ serum.

However, there are still numerous positive results of beneficial antioxidant effects during cancer chemotherapies and cancer cachexia pathogenesis [38,42].

### 3.1. Possible Mechanism of Action of Antioxidants

The concept of antioxidants is quite complex; in general, an antioxidant is a molecule or drug that hinders an oxidation reaction. Therefore, the definition of oxidation is a chemical process whereby electrons are lost during the reaction by the chemical species that are involved. These electrons are gained from a different chemical species, and this process is called reduction. Oxidation and reduction reactions occur coupled, and these processes are referred to as redox reactions [43]. These reactions are important for cell physiology; however, in some imbalance situations, they are harmful to the system and have deleterious effects. Oxygen is the terminal oxidant of the respiratory or electron transport chain [44]; on the one hand, it is essential for life, while, on the other, it causes various cellular damage during the production of reactive oxygen species (ROS), such as superoxide anion (O_2_^−^), hydrogen peroxide (H_2_O_2_), and hydroxyl radical (HO•). High ROS levels or a decrease in the cellular antioxidant capacity leads to cell oxidative stress, resulting in ROS-mediated damage of nucleic acids, proteins, and lipids. Oxidative stress is implicated in various disease states such as atherosclerosis, cancer, neurodegeneration, and aging [45]. It has been reported that ROS can interact with driver pathways to initiate signaling in a wide variety of cellular processes, such as proliferation and survival (MAP kinase, PI3 kinase, and PTEN), ROS homeostasis and antioxidant gene regulation (thioredoxin, peroxiredoxin, Ref-1, and Nrf-2), mitochondrial oxidative stress, apoptosis, aging, and DNA damage response [45,46,47].

Very recent data suggest that natural plant-derived antioxidants could have therapeutic properties by modulating microRNAs (miRNAs), a class of noncoding RNA, that are involved in inflammation and carcinogenesis and are deregulated in diverse tumors, including PCa [48]. These findings implicate that the use of antioxidants may be an attractive miRNA-mediated chemopreventive and therapeutic option in PCa.

### 3.2. ROS in Prostate Cancer

Growing evidence points out that PCa is closely associated with aging mechanisms, and high levels of ROS induced by aging activate several pathways which facilitate the onset, development, and progression of PCa [8]. Various sources of intracellular ROS are reported to contribute to the pathogenesis and the progression of PCa. Some of these high levels of reactive oxygen forms result from dysfunctional mitochondrial cellular respiration, the Warburg effect (altered glucose metabolism), overexpression of p66Shc related to age disorders, and the activation of enzymes such as NADPH oxidases, xanthine oxidases, and cytochrome P450. Furthermore, non-physiological ROS levels are associated with oxidative damage of proteins, lipids, and nucleic acids [49]. Other studies have shown that ROS production and oxidative stress led to androgen stimulation in androgen receptor (AR)-positive cells of PCa. In fact, androgens activate AR signaling driving the growth and metastasis, while simultaneously suppressing the apoptosis of PCa cells [50]. Epidemiological studies strongly suggest that a lower risk of cancer is associated with diets that indicated a high consumption of fruit and vegetables; thus, the active ingredients of these foods were tested to verify their preventive and anticancer properties, and to study their molecular mechanism of action [51].

Below, we discuss possible molecular mechanisms and interesting results obtained during recent years for a limited number of antioxidants, namely, vitamin E, lycopene, and green tea, which are among the most controversial and emerging ones in the field.

## 4. Vitamin E

Vitamin E is a fat-soluble vitamin belonging to the large family of four tocopherols and four tocotrienols; however, alpha-tocopherol is the only form of vitamin E used in humans as a supplement. Vitamin E is found in plant-based oils, nuts, seeds, fruits, and vegetables (Figure 1). The bioavailability of vitamin E metabolites has only been recently reported. Long-chain or short-chain carboxychromanols such as 13′-COOH, 11′-COOH, and terminal metabolite 3′-carboxychromanol (3′-COOH), also called 3′-carboxy-ethyl 6-hydroxy chroman (CEHC), seem to be the principal metabolites in the plasma [52]. The main role of vitamin E is to function as an antioxidant, scavenging free radicals and protecting against chronic diseases, such as heart and blood vessel disease [53].

During these last years, diverse population studies, clinical trials, and basic research reports have highlighted some benefits, as well as provided evidence against the intake of vitamin E in the limitation of PCa progression [54,55,56,57,58]. The most recent meta-analysis study on the association between antioxidants and prostate cancer association, comprising 18 studies, showed that higher dietary intakes of selenium, vitamin C, vitamin E, and β-carotene were significantly correlated with a reduced pancreatic cancer risk [59].

Recently, upon analyzing the clinical trial data after a long period (20 years), it was understood that the studies should be protracted, and the subjects recruited should be analyzed more homogeneously considering smokers and nonsmokers, as well as the type of prostate cancer, i.e., early-stage or advanced-stage cancer. Previous studies on vitamin E supplements and prostate cancer found the greatest benefit in men with more advanced smokers and cancers [54,55]. In the SELECT study, however, <10% of men were smokers, most had an early-stage cancer detected by prostate-specific antigen (PSA) blood tests, and the study was stopped earlier than expected as it did not appear to give no cancer or prostate cancer prevention results [56,60] (Table 1). Many early, low-grade prostate cancers identified by the PSA test do not become advanced cancers. Prostate cancer develops slowly, and any prostate cancer prevention study must be carried out for a long time. By discontinuing the SELECT study, there is no way to say whether vitamin E could have helped protect against prostate cancer. Very few cases in the SELECT study were of advanced prostate cancer, further limiting the interpretation of the results.

Even basic research is still debating the molecular mechanisms involved in the scavenger action of vitamin E, while trying to understand if there are side-effects that may instead have an inductive role of the tumor. Interestingly, a model of benign (primary), premalignant (RWPE-1), and malignant (LNCaP) prostate epithelial organoids showed that vitamin E supplementation decreased proliferation and induced apoptosis in cancer organoids, while it had no effect on the benign organoids. In contrast, vitamin E enhanced cell proliferation in the premalignant organoids via fatty acid oxidation in a manner that recapitulated the SELECT results [61]. Other papers have reported how the organoid model, in comparison to 2D cultures, is a robust 3D in vitro system which may elucidate crucial mechanisms of oncogenesis and develop novel targeting strategies by decreasing the discrepancies found between results in cell lines and clinical trials [62]. Supporting the emerging data on the nonbeneficial effects of vitamin E intake on healthy subjects, it was recently reported that vitamin E in the human prostate epithelial RWPE-1 cell line and in the rat model significantly upregulated the expression of various phase-I activating cytochrome P450 (CYP) enzymes, including activators of polycyclic aromatic hydrocarbons (PAHs), leading to supraphysiological levels of ROS [63]. Furthermore, the authors showed that vitamin E caused DNA damage, which promoted cell transformation induced by an increase in the cellular PAH prototype benzo[a]pyrene. In a recent study, the associations of multivitamin use with the predicted risk of prostate cancer were compared using observed biopsy data. The obtained results initially correlated an increased risk of high-grade prostate cancer with current and long-term use of multivitamins. Subsequently, further observations showed that the associations of multivitamin use with prostate cancer were attenuated and not statistically significant [64].

Some authors showed that δ-tocotrienol (δ-TT), a form associated with the chemosensibilization of PCa to gemcitabine activity, was involved in the Warburg effect, a novel hallmark of all tumors [65,66]. The results showed that δ-TT inhibited glucose uptake and lactate production in PTEN-deficient LNCaP and PC3 PCa cells, by specifically decreasing hexokinase 2 (HK2) expression, with the concomitant inhibition of the Akt pathway causing a decrease in cell growth [66]. Interestingly δ-TT synergized with metformin in inducing PCa cell death. Metformin is a hypoglycemic agent, used in diabetic patients but repositioned in the therapy of other diseases such as breast cancer prevention [67]. These results highlighted the crucial role of the metabolic phenotype of PCa in δ-TT-mediated cytotoxicity.

Some papers have reported that different forms of vitamin E and, therefore, different metabolites exert different effects on cell proliferation or death in different types of cancer or in the same type of cancer at different stages [68,69]. These multiple aspects obviously complicate the scenario but undoubtedly shed light on how difficult it is to interpret the results, which require an investigative approach that is not only clinical and epidemiological, but also chemical and molecular. An interesting and recent study showed that β-tocotrienol (β-T3), an isomer of vitamin E, inhibited PD-L1 expression and PD-L1-mediated tumor-promoting function in both in vitro and in vivo lung and prostate models. This mechanism inactivated the JAK2/STAT3 pathway while increasing the immune response, contributing to the inhibition of several oncogenic activities tested [70].

Furthermore, the combined effects of treatments in which different molecular mechanisms could overlap were underestimated in several studies as they could counteract the beneficial effects of the singular treatment. The SELECT study discussed earlier sought to determine whether selenium and/or vitamin E could prevent prostate cancer and other diseases with little or no toxicity in relatively healthy men [60]. As discussed, selenium and vitamin E, alone or in combination at the doses and formulations used, did not prevent prostate cancer in the relatively healthy men considered. The hypotheses generated by the scientific community to explain the null results centered on the agent formulation, chosen dosage, the cohort, and the study design. Some groups tried to reproduce in the laboratory the conditions proposed by SELECT on prostate tumor lines; however, neither selenium (l-selenomethionine) nor vitamin E alone or together showed antitumor activity [71,72].

A clearer view of selenium’s biological activities will aid researchers in choosing the appropriate doses and formulations of future agents. Work is continuing on selenium and its ability to antagonize carcinogenesis. One promising approach is the ongoing characterization of selenium’s anti-DNA damage activities.

## 5. Lycopene

Lycopene, the red pigment of tomatoes and other fruits (Figure 1), is the most abundant carotenoid in tomatoes; however, due to its chemical structure, it differs from other carotenes such as α-carotene and β-carotene [73]. Several studies have reported that red tomato is the best food to deliver lycopene in human metabolism.

Bioavailability of lycopene is greatly affected by dietary composition because it is a lipid-soluble compound. The uptake of lycopene with fatty food increases its bioavailability, resulting in higher blood carotenoid levels [74]. The study of bioavailability of lycopene in human nutrition reported that the all-trans isomer predominates in the main dietary source of lycopene (i.e., tomatoes), but the isomer which is then isolated into blood, plasma, and tissues has the cis conformation. The processing of tomatoes by heating probably converts the all-trans lycopene into cis isomers [74]. The conformation change could lead to a decrease in the activity of catabolic reactions, which are selected in nature to recognize the trans-isomer conformation.

Mouse models of prostate cancer have provided the opportunity for many groups to evaluate the effects of lycopene supplementation and study its molecular mechanisms. Pure lycopene intake showed anticancer activity in most studies, although the results varied by model system and were sometimes conflicting, suggesting that the impact of lycopene consumption may depend on the dose, duration, and different stages of tumorigenesis [75].

A high number of papers have reported diverse molecular mechanisms via which the active metabolites of lycopene are capable of counteracting cancer cell growth and oncogenic functions, which are reflected in the transcriptional program change of genes involved in migration and invasion [76,77]. A very interesting aspect of recent research is that lycopene may play a role in the correlation between inflammation and cancer. It is now accepted that cancer-associated inflammation can arise as a result of tumor hyperproliferation or that it is the basis for the development of cancer, creating a favorable microenvironment [78]. These studies pointed out that inflammatory cytokines such as tumor necrosis factor (TNF), IL-1 and IL-6, are induced by hypoxia and inflammation of the tumor microenvironment (TME) [79]. They are able to stimulate the JNK and NF-κB pathways, promoting an increase in aberrant proliferation and the inhibition of apoptosis. These activities promote ROS and free-radical generation in the TME [80]. Interestingly, several research groups have documented that lycopene could suppress proinflammatory cytokines such as IL-1, IL-1β, IL-6, and TNF-α in several tumor models, including PCa, by downregulating their production and, thus, preventing the inflammatory state [81,82].

Treatment of prostate cell lines with lycopene has shown the modulation of some pathways involved in the arrest of proliferation and in apoptosis by affecting driver gene expression such as Bax, Bcl-2/IGFBP-3, uPAR, TP53, Cyclin-D1, and Nrf-2 [83,84,85,86].

Several trials have tried to correlate the effects of lycopene consumption with the progression of prostate cancer. In a meta-analysis study on 26 clinical trials available up to 2014 (17,517 PCa patients from 563,299 participants), Chen and colleagues reported a trend in the inverse association between high lycopene intake consumption and lower PCa risk (*p* = 0.078) [87].

Very recently, some phase I and II studies were conducted to investigate the metabolites of lycopene in the plasma of patients and the activity of lycopene in association with some chemotherapeutic agents used in PCa therapies [88,89,90] (Table 1). The results described were not statistically significant, e.g., due to small sample sizes; however, the trend is very encouraging, and the combination of chemotherapies and lycopene merits further study.

## 6. Green Tea

Green tea for human consumption is extracted from the leaves of Camellia sinensis (Theaceae family) (Figure 1) and is widely consumed in Asian countries [91]. In recent years, its consumption has also spread to European countries, as it has assumed great importance as an antioxidant, anti-inflammatory, and antibacterial agent [92]. The active ingredients contained in green tea constitute a class of polyphenols called catechins (GTCs) which give the characteristic bitter taste to the drink obtained by infusing the leaves. The GTCs present in green tea are epicatechin (EC), epigallocatechin-3-gallate (EGCG), epigallocatechin (EGC), and epicatechin-3-gallate (ECG) [93]. Several in vitro and in vivo studies have shown that EGCG is the molecule most involved in the pathways underlying prostate carcinogenesis, revealing itself today as one of the most promising and most studied antioxidants in pathologies of the reproductive system [93,94], although a very recent study reported that green tea infusion does not affect tumor growth in xenograft and allograft models [95].

Recently, a very interesting study evaluated the bioavailability and pharmacokinetic profile of epigallocatechin gallate (EGCG) alone or in combination with different food supplements in a cohort of healthy volunteers, to provide a consensus on the most appropriate concentration to obtain the maximum therapeutic effects [96].

A fair number of studies from a few years ago showed sometimes significant anticorrelations between the consumption of green tea and the risk of developing PCa, and other times trends, which are very interesting [97,98,99] (Table 1). At the molecular level, the bulk of the in vitro experiments were performed on prostate cancer DU-145, LNCaP, and PC-3 cells. Treatment with EGCG, ECG, and ECG, but not EC, has been reported to have a significant inhibitory effect on cell proliferation by increasing p21, KIP1/p27, INK4a/p16, and INK4c/p18 expression while decreasing the expression of cyclin D1, cyclin E, cyclin-dependent kinase 2 (CDK2), CDK4, and CDK6 [100,101]. EGCG was also able to induce cell death [102]. EGCG has been shown to lead to increased levels of TP53 mRNA in LNCaP cells that express wildtype p53 protein, leading to cell-cycle arrest and apoptosis [103]. However, EGCG treatment did not have the same effect on DU145 cells [103], probably because these cells express a mutated p53 protein Val274Phe with gain-of-function oncogenic activity, whereby the transcriptional pathway of wildtype p53 is no longer functional. This allows reflecting on the fact that some pathways that induce green tea are closely connected with the response to damage.

One last aspect that has made green tea a very interesting antioxidant in recent years is its ability to modulate microRNAs (miRs) in different types of cancer, including PCa [104,105]. MiRs are a group of noncoding, single-stranded RNAs (~22 nt in length) involved in the tumorigenesis of various cancers, including PCa, by implementing different molecular strategies [106]. Increasing studies have reported that miR-93 plays numerous roles in both physiological and pathological mechanisms; in particular, it can act both as an oncomiR and as a tumor suppressor depending on the type of tumor [107]. In PCa, it was demonstrated to act as an oncomiR by increasing tumor cell growth, invasion, and migration [107]. Recently, it was shown that miR-93 induced PCa proliferation and migration by activating AKT/ERK signaling via downregulation of disabled homolog 2 (DAB2) protein expression [108]. Green tea has been shown to decrease miR-93 expression with the concomitant upregulation of DAB2 expression, inactivating AKT and ERK pathways [108].

Interestingly, the antitumor activity of green tea was evaluated in the LNCaP PCa 3D cell model through the induction of miR-181a expression [109]. In PCa, miR-181a acts as tumor suppressor, and the green tea treatment rescued its expression while causing the induction of the apoptotic pathway via BCL-2 and BAX upregulation [109].

These results related to the ability of green tea to modulate the expression of miRNAs should give impetus to new studies designing the massive screening of noncoding RNAs in ongoing clinical trials to search for circulating blood markers that are correlated with the antioxidant activity in the patient. For example, an interesting study considered obese women over 18 years old who had no comorbidities instructed to take green tea as an integrator or to take a placebo [110] (Table 1). From the analysis of blood samples at different times, it emerged that supplementation attenuated inflammatory and oxidative stress biomarkers related to high-fat and saturated meals through its ability to modulate the expression of some miRNAs circulating in the blood [110].

## 7. Clinical Studies

Several observational studies, some dating back almost 20 years and others more recent, have studied the effect of specific dietary antioxidants on the incidence and progression of prostate cancer [111,112]. The most numerous studies carried out on humans have focused on carotenoids, particularly beta-carotene and lycopene, on vitamins E and C, on food sources of various phenolic substances such as coffee and tea, and on flavonoids. Overall, many of these studies were undefined and inconclusive with respect to actual benefits, with various antioxidants affecting prostate cancer risk differently.

To date (December 2022), 81 clinical studies are reported on ClinicalTrials.gov (https://clinicaltrials.gov/ct2/home; keywords were “prostate cancer” and “antioxidant” accessed on 1 December 2022), 20 of which are complete with the results available, with the aim of exploring the effect of antioxidant treatment on the attenuation of prostate cancer development or preventing the onset of prostatic hyperplasia or prostate cancer in healthy men with very long follow-ups.

Overall, in these studies, the researchers introduced vitamin E, vitamin C, vitamin D, carotenoids, selenium, and green tea extract through the diet by integrating foods rich in these compounds or orally with concentrated supplements of the active ingredient. Some trials interestingly evaluated the effect of nutraceutical antioxidant agents coupled with chemotherapy versus chemotherapy alone (NCT01882985 with results; NCT01949519; NCT05501548), while other studies are recruiting to test juices and plant extracts such as green tea extract (NCT00744549; NCT01912820; NCT00003367; NCT01105338; NCT00459407; NCT00253643 with results), pomegranate juice (NCT00732043; NCT00731848), acai juice (NCT01521949 with results), silybin extract (NCT00487721 with results), and grape seed extract (NCT03087903).

Among the abovementioned studies reporting results, the NCT00416325 phase II trial was interesting, as lycopene carotenoid is a potent antioxidant [113]. It has been reported that lycopene could exert its antioxidant and anti-inflammatory effects through modulation of the NF-κB pathway in various tumors, including prostate disease [82,114,115]. The main aim of the study was to compare, in a timeframe of 6 months, the effects of a lycopene supplement from tomatoes versus a placebo in men with high-grade prostatic intraepithelial neoplasia (HGPIN). The authors evaluated through biopsy the expression of proteins marking the status of proliferation, differentiation, cell regulation, and apoptosis in high-risk tissue and changes in serum biomarkers (PSA, IGF-1, and IGFBP3). The consumption of a lycopene-rich tomato extract did not lead to a significant effect on proliferation or cell-cycle inhibition, despite a substantial increase in serum lycopene, with no effect of on circulating levels of PSA, IGF-1, or IGFBP3 [116]. In contrast, a previous trial reported that levels in serum lycopene were inversely associated with serum IGF-1 levels [117]. The limitation of this trial may be its relatively small size of the patient cohort and the restricted statistical power. However, the anticancerogenic properties of lycopene are associated with several mechanisms, and there are many variables to consider, such as the markers to be considered, the detection technologies used, and the timeframe of data collection. For example, lycopene could modulate noncoding RNAs [118]; however, according to our research, there are no clinical studies evaluating the presence of noncoding RNA in tissues or circulating in patients treated with lycopene. On the other hand, a recent meta-analysis review including 42 papers (692,012 participants and 43,851 prostate cancer cases) reported that dietary lycopene intake and circulating concentration were significantly related to a lower risk of prostate cancer [119]. The conclusions that can be drawn is that future trials could benefit from a longer duration and the use of alternate biomarker endpoints.

A very recent phase II clinical trial (NCT01882985) investigated the activity of docetaxel chemotherapy plus lycopene in advanced castrate-resistant adenocarcinoma of the prostate with a ≥50% reduction in PSA, median time to PSA progression, duration of response, and overall survival as the principal endpoints [89] (Table 1). The reported results suggested that docetaxel plus lycopene had a favorable activity in patients with metastatic castration-resistant prostate cancer, in terms of patient tolerability to treatment and reduction of adverse effects. Furthermore, lycopene had a synergistic activity with docetaxel through the downregulation of the inhibition of IGF-I signaling and the increase in survival [89]. Therefore, the coupling of the chemotherapy treatment together with high-dose lycopene may have very promising future applications in more in-depth clinical trials.

Plant derivatives have been studied as therapies for prostate cancer as a function of their antioxidant and anti-inflammatory abilities and high tolerability. Euterpe oleracea (açaí) is abundant in South and Central America, and the acai berry is rich in phytochemicals with anticarcinogenic and chemopreventive activities as observed in many experimental models of tumors, showing reduced tumor cell proliferation, multiplicity, and size [120]. In a quite recent phase II clinical trial, a cohort of men with asymptomatic or minimally symptomatic prostate cancer with rising PSA were treated with acai juice product (NCT01521949 [121]) (Table 1). The research did not accomplish its primary endpoint of PSA response >50%; however, the authors reported a slight decrease in PSA levels and a slowing of doubling time in many patients. Notably, one patient had a PSA response within the study time period. Therefore, these are encouraging data supporting the hypothesis that the use of a low-risk natural product such as acai juice with antioxidant properties could exert anticancer effects. As commented above, this study might also suggest that additional genomic predictive markers should be considered, such as the genetic polymorphism associated with response and/or modulation of noncoding RNAs.

Silybin or silibinin is another interesting antioxidant widely documented. It is a flavonolignan isolated from milk thistle seeds, extensively studied for its antioxidant, hepatoprotective, and anticancer properties [122]. Silibinin has demonstrated anticancer activity for several different cancer types, including prostate cancer [123]. The patients with localized prostate cancer who participated in the phase II clinical trial NCT00487721 received oral administration of silybin phytosome three times a day from enrollment in the study until the time of their prostatectomy. The scientists analyzed the effect and the levels of silybin phytosome by analyzing blood and urine samples at the start and completion of the trial in addition to prostate tissue from the surgery. Silybin was found at high concentrations in blood transiently, and low levels were seen in prostate tissue. However, tissue penetration was considered low, due to the molecule’s short half-life, the brief duration of therapy in this study, or an active process removing silibinin from the prostate. All these aspects should be fine-tuned before a phase III clinical trial can be initiated [124] (Table 1).

**Table 1 antioxidants-12-00289-t001:** Clinical trials that recently investigated the role of antioxidants in the PCa, as discussed in the review.

ID Trial; Phase	Purpose	Antioxidant	Reference
NCT00006392 (SELECT); Phase III, 4 arms	Randomized phase III trial to determine the effectiveness of selenium and vitamin E, either alone or together, in preventing prostate cancer.	Vitamin E, selenium	[53,56]
NCT00433797; Phase I/II, 3 arms	Outcomes included serum PSA kinetics, as well as biomarkers of inflammation, antioxidant status, oxidative stress, and oxidative damage in blood cells, plasma, urine, and prostate tissues.	Tomato	[88]
NCT01882985; Phase II, 1 arm	This phase II trial evaluated the impact of giving docetaxel together with lycopene supplements in treating patients with hormone-resistant prostate cancer not previously treated with chemotherapy.	Docetaxel, lycopene	[89]
Case–control study	Study of the association of circulating carotenoids and retinol with intraprostatic inflammation in benign tissue.	Lycopene, carotenoids, retinoids	[90]
Case–control study, 2 arms	Investigation whether green tea usually consumption had an etiological association with prostate cancer development.	Green tea	[97]
Phase II	Evaluation of the efficacy of green tea catechins for chemoprevention of PCa in patients with high-grade prostate intraepithelial neoplasia.	Green tea	[99]
NCT00685516; Phase II	Study of the effect of green tea and black tea consumption on biomarkers related to prostate cancer development and progression.	Green tea, decaffeinated black tea	[98]
Phase II, 2 arms	Evaluation of green tea supplementation on the ability to attenuate inflammatory and oxidative stress biomarkers induced by high-fat, high-saturated meals in obese women, and to modulate circulating microRNA (miRNA) expression.	Green tea	[110]
NCT00416325; Phase I	Study of the side-effects and best dose of lycopene in preventing prostate cancer in patients who are at high risk of developing prostate cancer.	Lycopene	[113]
NCT01521949; Phase II	Study of the effect and tolerability of the acai berry in in patients with biochemically recurrent prostate cancer with a primary endpoint of prostate-specific antigen (PSA) response.	Acai juice	[121]
NCT00487721; Phase II, 2 arms	Investigation whether the silibinin levels were detectable in human fluid and tissue samples men affected by adenocarcinoma of the prostate.	Silibin phytosome	[124]

## 8. Conclusions

Antioxidants are natural or artificial molecules that can scavenge free radicals and prevent oxidative DNA damage. Although, in the last 20 years, the scientific literature has seen a robust increase in the number of papers that have dealt with the medical aspects and physical/clinical parameters regarding the uptake of antioxidant supplements, the molecular mechanisms that support many of these are not sufficient to explain some discrepancies.

Many clinical trials and population studies have been conducted without a specific therapeutic regimen due to the lack of guidelines. Each study has a protocol which is discussed by the authors on the basis of the results. The meta-analyses found the lack of specific guidelines for the absorption of these phytochemicals to be the weakness of these studies. This fact determines that much evidence remains inconclusive, sometimes creating discrepancies between in vitro and clinical results on the ability of antioxidants to prevent the onset of PCa. For these reasons, the protective mechanistic pathways of antioxidants remain to be fully dissected.

On the risk of developing PCa in healthy patients, the effect of the absorption of some antioxidants remains inconclusive. According to in vitro experiments, the results are better on pancreatic cancer tumor lines, as a reduction in proliferation and migration capacity have been documented.

Some authors, who conducted extensive meta-analyses, revealed many heterogeneous factors in the populations of healthy individuals and PCa patients that could affect the final data. Often, the too low number of individuals of these trials and the early interruption of data recording, because of seemingly no promising results regarding the starting hypothesis, have invalidated the global analysis. It is no small fact that the ingestion of particular foods in the diet or the use of medicines can often counteract the overall effect of the antioxidant uptake under study.

Recent studies have shown that genetic alterations, as happens during carcinogenesis, as well as polymorphisms or epigenetic modifications of DNA, modify the response and the protective effect of antioxidant agents [125,126]. The biological analysis approach across multiple levels that combines multi-omics techniques such as transcriptomics, proteomics, genomics, epigenetics, and metabolomics could allow the identification of crucial molecular pathways in diverse and well-characterized groups of patients which are modulated by the uptake of antioxidant supplements. Moreover, these multi-omics studies could incentivize the design of nutraceutical-driven precision medicine strategies to develop PCa patient-targeted therapies and control chemotherapy-associated toxicities, thereby decreasing the acquired resistance to chemotherapy in both localized and advanced-stage PCa.

The identification of miRNAs linked to specific signaling pathways that are related to PCa progression and metastasis could also be important for providing novel antioxidants as therapeutic opportunities. These miRNAs are easily detectable in the blood or urine by “liquid biopsy”, representing attractive biomarkers for diagnosis, as well as the monitoring of therapy response and prognosis in prostate cancer.

## Figures and Tables

**Figure 1 antioxidants-12-00289-f001:**
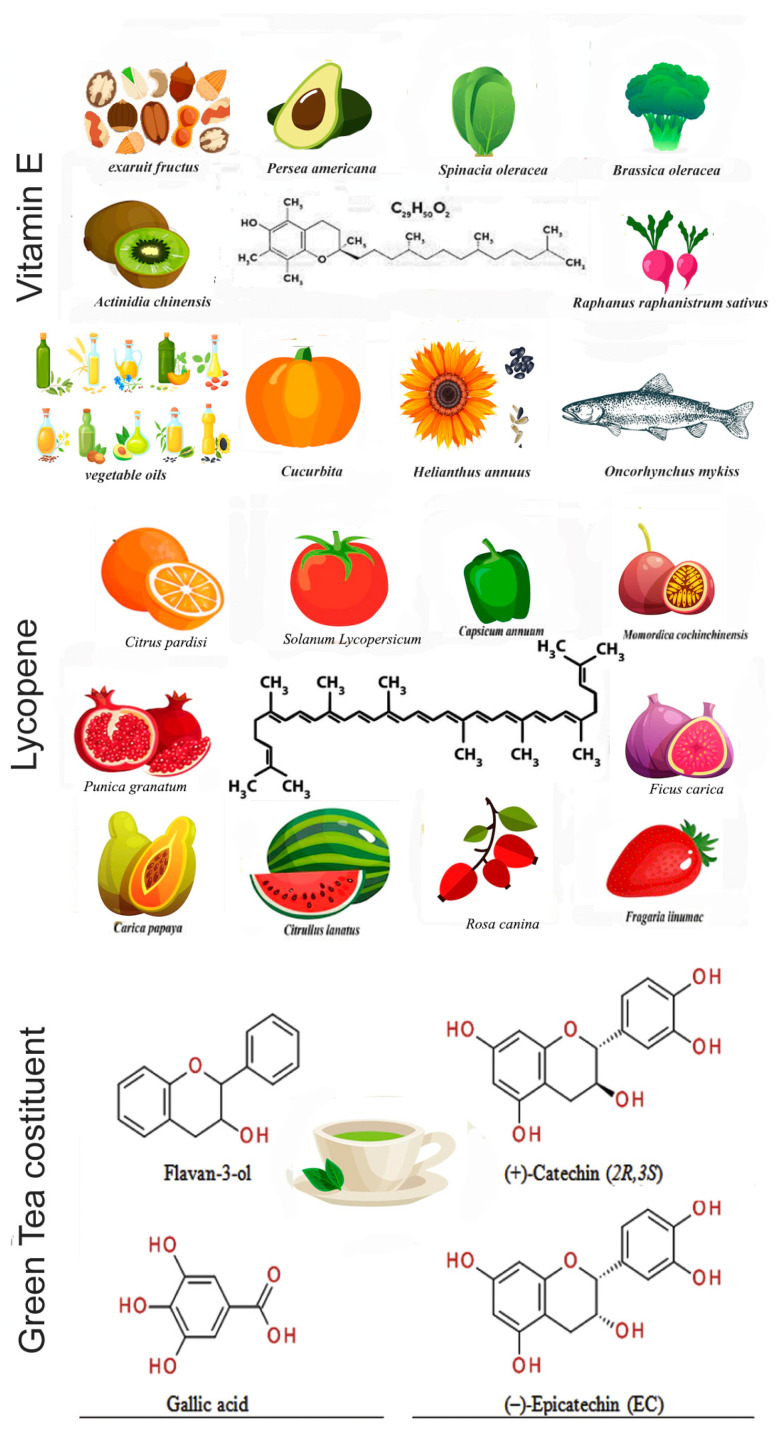
Principal main foods as a source of vitamin E and lycopene, along with green tea constituents (images of sources of antioxidants were taken from https://it.freepik.com) accessed on 1 December 2022.

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
