# Peer review of "Novel Insights into the Role of the Antioxidants in Prostate Pathology"

_antioxidants, 2023, doi:10.3390/antiox12020289_

Round 1
Reviewer 1 Report
The manuscript- “Novel insights into the role of the antioxidants in prostate pathology” by Rago and Agostino attempts to provide insights into the health beneficial roles of antioxidants in prostate tumor. Although this manuscript outlines literature review on advancement made on chemoprevention by dietary antioxidants, I have the following comments.
1. There is no insight into recommended dosing regimen of antioxidants for potential prostate cancer chemoprevention.
2. The authors should provide updated literature review on the bioavailability issue of the antioxidants. It is unclear whether antioxidant effects of compounds are due to original chemical structures or metabolized forms, given their relatively low bioavailability in vivo. In addition, there are missing information on variability of these antioxidant compounds in dietary sources.
Author Response
Reviewer 1
The manuscript- “Novel insights into the role of the antioxidants in prostate pathology” by Rago and Agostino attempts to provide insights into the health beneficial roles of antioxidants in prostate tumor. Although this manuscript outlines literature review on advancement made on chemoprevention by dietary antioxidants, I have the following comments.
- There is no insight into recommended dosing regimen of antioxidants for potential prostate cancer chemoprevention.
Reply: We thank the reviewer for the right observation.
Among the clinical studies that we have discussed, some reported the dosage of the purified and ingested active ingredient according to a precise prescription. For example, in the NCT00487721 trial (ref. 115, 116), the participants took Silibin-Phytosome 13 grams daily, in three divided doses for 2-10 weeks. However, this, like other studies, did not have guidelines that can guide the adoption of a specific regimen. Each study reported a protocol that was further discussed by the authors basing on the results. In fact, diverse meta-analysis studies concluded that the lack of specific guidelines about the uptake of these phytocompounds was the weak point of these studies. This fact determines that much evidence remains inconclusive, sometimes creating discrepancies about the ability to prevent the onset of PCa. For example, we discussed the inconsistencies in the SELECT study NCT00006392 (REF 53, 56) where the men took 400 International Units (IU) of vitamin E, a dose higher than the dose used in the previous studies (50 mg/day) (Sesso HD, Buring JE, Christen WG, et al. Vitamins E and C in the prevention of cardiovascular disease in men: the Physicians' Health Study II randomized controlled trial. JAMA. 2008; 300(18):2123-33).
Another group of studies evaluated the consumption of fruit and vegetables within a cohort of individuals. It has been demonstrated by in vitro experiments on cell lines that fruits and vegetables possess molecules with a PCa chemopreventive potential, but also in this case their uptake in the food has not been clearly correlated with a reduction in the risk of PCa development in the men.
In the revised version of the manuscript, we inserted these comments in the Conclusions.
- The authors should provide updated literature review on the bioavailability issue of the antioxidants. It is unclear whether antioxidant effects of compounds are due to original chemical structures or metabolized forms, given their relatively low bioavailability in vivo. In addition, there are missing information on variability of these antioxidant compounds in dietary sources.
Reply: We thank the reviewer for this observation.
Accordingly, recent evidence suggested that the pharmacokinetics and metabolism of bioactive compounds are crucial to understanding their role and function in PCa and in other disease. However, the exact mechanisms of action, effects, and bioavailability are still not fully recognized.
We have integrated this information into the text in the respective parts of interest.
New references
- Drużyńska, B.; Wołosiak, R.; Grzebalska, M.; Majewska, E.; Ciecierska, M.; Worobiej, E. Comparison of the Content of Selected Bioactive Components and Antiradical Properties in Yoghurts Enriched with Chia Seeds (Salvia hispanica L.) and Chia Seeds Soaked in Apple Juice. Antioxidants 2021, 10, 1989.
- Mohd Zaffarin AS, Ng SF, Ng MH, Hassan H, Alias E. Pharmacology and Pharmacokinetics of Vitamin E: Nanoformulations to Enhance Bioavailability. Int J Nanomedicine. 2020 Dec 8;15:9961-9974. doi: 10.2147/IJN.S276355.
- Caponio, G.; Noviello, M.; Calabrese, F.; Gambacorta, G.; Giannelli, G.; De Angelis, M. Effects of Grape Pomace Polyphenols and In Vitro Gastrointestinal Digestion on Antimicrobial Activity: Recovery of Bioactive Compounds. Antioxidants 2022, 11, 567.
- Madia VN, De Vita D, Ialongo D, et al. Recent Advances in Recovery of Lycopene from Tomato Waste: A Potent Antioxidant with Endless Benefits. Molecules. 2021;26(15):4495. Published 2021 Jul 26. doi:10.3390/molecules26154495
- Andreu-Fernández V, Almeida Toledano L, Pizarro N, et al. Bioavailability of Epigallocatechin Gallate Administered With Different Nutritional Strategies in Healthy Volunteers. Antioxidants (Basel). 2020;9(5):440. Published 2020 May 19. doi:10.3390/antiox9050440

Reviewer 2 Report
Very good article that combines medicine with nutritional aspects in a very appropriate way.
Although it has an updated bibliography, it may be interesting to introduce another article from 2022.
For example:
Cancer Epidemiol Biomarkers Prev 2022 Nov 2;31(11):2063-2069.
doi: 10.1158/1055-9965.EPI-22-0453.
Differential Biopsy Patterns Influence Associations between Multivitamin Use and Prostate Cancer Risk in the Selenium and Vitamin E Cancer Prevention Trial
Jeannette M Schenk , Cathee Till, Marian L Neuhouser, Phyllis J Goodman , M Scott Lucia, Ian M Thompson, Catherine M Tangen
Prostate 2022 Aug;82(11):1117-1124.
doi: 10.1002/pros.24364. Epub 2022 Apr 29.
Effects of green tea on prostate carcinogenesis in rat models and a human prostate cancer xenograft model
, ,
Cancer Prev Res (Phila) 2022 Apr 1;15(4):233-245.
doi: 10.1158/1940-6207.CAPR-21-0508.
δ-Tocotrienol is the Most Potent Vitamin E Form in Inhibiting Prostate Cancer Cell Growth and Inhibits Prostate Carcinogenesis in Ptenp-/- Mice
, , ,
Author Response
Reviewer 2
Very good article that combines medicine with nutritional aspects in a very appropriate way. Although it has an updated bibliography, it may be interesting to introduce another article from 2022.
For example:
-Jeannette M Schenk , Cathee Till, Marian L Neuhouser, Phyllis J Goodman , M Scott Lucia, Ian M Thompson, Catherine M Tangen. Differential Biopsy Patterns Influence Associations between Multivitamin Use and Prostate Cancer Risk in the Selenium and Vitamin E Cancer Prevention Trial. Cancer Epidemiol Biomarkers Prev 2022 Nov 2;31(11):2063-2069. doi: 10.1158/1055-9965.EPI-22-0453.
-Maarten C Bosland , Lori Horton, Mark S Condon Effects of green tea on prostate carcinogenesis in rat models and a human prostate cancer xenograft modelProstate 2022 Aug;82(11):1117-1124. doi: 10.1002/pros.24364. Epub 2022 Apr 29.
-Hong Wang, William Yan, Yuhai Sun, Chung S Yang δ-Tocotrienol is the Most Potent Vitamin E Form in Inhibiting Prostate Cancer Cell Growth and Inhibits Prostate Carcinogenesis in Ptenp-/- Mice.Cancer Prev Res (Phila) 2022 Apr 1;15(4):233-245.doi: 10.1158/1940-6207.CAPR-21-0508.
Reply: The authors thank the reviewer for enthusiastic comments and, as suggested, have added the recent references suggested by rev2 into the text.
- -Hong Wang, William Yan, Yuhai Sun, Chung S Yang δ-Tocotrienol is the Most Potent Vitamin E Form in Inhibiting Prostate Cancer Cell Growth and Inhibits Prostate Carcinogenesis in Ptenp-/- Mice.Cancer Prev Res (Phila) 2022 Apr 1;15(4):233-245.doi: 10.1158/1940-6207.CAPR-21-0508.
- Jeannette M Schenk , Cathee Till, Marian L Neuhouser, Phyllis J Goodman , M Scott Lucia, Ian M Thompson, Catherine M Tangen. Differential Biopsy Patterns Influence Associations between Multivitamin Use and Prostate Cancer Risk in the Selenium and Vitamin E Cancer Prevention Trial. Cancer Epidemiol Biomarkers Prev 2022 Nov 2;31(11):2063-2069. doi: 10.1158/1055-9965.EPI-22-0453.
- Maarten C Bosland , Lori Horton, Mark S Condon Effects of green tea on prostate carcinogenesis in rat models and a human prostate cancer xenograft modelProstate 2022 Aug;82(11):1117-1124. doi: 10.1002/pros.24364. Epub 2022 Apr 29.
